# Non-Haemodynamic Mechanisms Underlying Hypertension-Associated Damage in Target Kidney Components

**DOI:** 10.3390/ijms24119422

**Published:** 2023-05-29

**Authors:** Elisa Russo, Elisabetta Bussalino, Lucia Macciò, Daniela Verzola, Michela Saio, Pasquale Esposito, Giovanna Leoncini, Roberto Pontremoli, Francesca Viazzi

**Affiliations:** 1U.O.C. Nefrologia e Dialisi, Ospedale San Luca, 55100 Lucca, Italy; elisa24russo@gmail.com; 2IRCCS Ospedale Policlinico San Martino, 16132 Genova, Italy; betta.bussalino@gmail.com (E.B.); lucia.maccio@hotmail.it (L.M.); daverz@libero.it (D.V.); pasqualeesposito@hotmail.com (P.E.); giovanna.leoncini@unige.it (G.L.); roberto.pontremoli@unige.it (R.P.); 3Department of Internal Medicine, University of Genova, 16132 Genova, Italy; 4S.S.D. Nefrologia e Dialisi, Ospedale di Sestri Levante, 16124 Genova, Italy; michela.saio@virgilio.it

**Keywords:** arterial hypertension, kidney disease, RAAS, uric acid, treatment, novel therapeutic targets

## Abstract

Arterial hypertension (AH) is a global challenge that greatly impacts cardiovascular morbidity and mortality worldwide. AH is a major risk factor for the development and progression of kidney disease. Several antihypertensive treatment options are already available to counteract the progression of kidney disease. Despite the implementation of the clinical use of renin–angiotensin aldosterone system (RAAS) inhibitors, gliflozins, endothelin receptor antagonists, and their combination, the kidney damage associated with AH is far from being resolved. Fortunately, recent studies on the molecular mechanisms of AH-induced kidney damage have identified novel potential therapeutic targets. Several pathophysiologic pathways have been shown to play a key role in AH-induced kidney damage, including inappropriate tissue activation of the RAAS and immunity system, leading to oxidative stress and inflammation. Moreover, the intracellular effects of increased uric acid and cell phenotype transition showed their link with changes in kidney structure in the early phase of AH. Emerging therapies targeting novel disease mechanisms could provide powerful approaches for hypertensive nephropathy management in the future. In this review, we would like to focus on the interactions of pathways linking the molecular consequences of AH to kidney damage, suggesting how old and new therapies could aim to protect the kidney.

## 1. Introduction

Arterial hypertension (AH) is the most common chronic disorder worldwide, representing the main prevalent modifiable risk factor associated with cardiovascular morbidity and mortality and the second most frequent cause of chronic kidney disease (CKD) [1,2]. AH has been shown to trigger molecular and histopathological changes leading to “benign” or “malignant” nephrosclerosis [3], and to a wider spectrum of hypertension-induced kidney damage that has been largely investigated in recent years.

Chronic high blood pressure (BP) generates mechanical damage along the vascular system, heart, and kidneys, which are the main organs affected by this condition. The pathogenetic determinants of hypertensive kidney damage can be attributed to the systemic BP load, the degree in which this load is transmitted to the kidney vascular bed, and the susceptibility to the barotrauma of the local tissue [4]. Increased albumin excretion rate is the earliest clinical manifestation of these balance games. Interestingly, BP behaviour and BP variability have shown a significant impact on kidney damage both in hypertensive and normotensive subjects [5,6], suggesting that excessive BP variability can produce, per se, end organ damages even in non-hypertension. Thus, it imprints an increased oscillatory shear stress to the vascular endothelium, potentially contributing to early atherosclerosis more than steady blood flow [7].

In the kidney, BP elevation produces damage at multiple levels: glomerular, tubule-interstitial, and vascular. In addition to mechanical stress, AH induces oxidative stress, chronic inflammation, and activation of the reparative mechanisms leading to kidney damage, mainly due to nephrosclerosis and fibrosis [8].

Recently, the paradigm for AH management has shifted toward a multi-factorial, organ-protection-cantered approach, from the heart to the kidney [9]. However, patients with AH remain at high risk of developing renal and cardiovascular complications. For this reason, there is a great need to identify the best target for therapeutic strategies to optimize kidney and cardiovascular protection.

This narrative review focuses on the biological mechanisms of AH involving kidney structures, in search of emerging therapies targeting novel mechanisms, to be useful for future research that will face new promising therapeutic approaches, as shown in Table 1.

## 2. Mechanisms Underlying Kidney Injury in Arterial Hypertension

### 2.1. Renin–Angiotensin–Aldosterone System

The renin–angiotensin–aldosterone system (RAAS) is one of the most critical hormonal mechanisms in controlling hemodynamic stability by regulating BP, fluid volume, and electrolyte balance. Because of this, an alteration in any of the molecules that compose RAAS might contribute to developing AH. Clinical trials on RAAS inhibitors (RAAS-is) have largely demonstrated that this intervention lowers morbidity/mortality, particularly in diabetic [10] and CKD patients [11]. Besides lowering BP, their beneficial effects are attributed to the inhibition of the undesired Angiotensin II Receptor Type-1 (AT1R) stimulation and subsequent reduction in vascular tone, aldosterone, vasopressin, and catecholamine release; inhibition of inflammation; and attenuation of cell growth [12,13,14]. Therefore, RAAS-is are considered first-line therapy for hypertensive patients with CKD or those at risk of kidney disease or those with diabetes [15].

Many trials have shown a new class of anti-diabetic drugs (namely sodium–glucose cotransport 2 inhibitors (SGLT2is), to be safe, effective, and capable of providing additional CV and renal benefits beyond their glucose-lowering effect. Moreover, the combined use of RAAS-is with SGLT2-is seems to have been proven to be an effective weapon to counteract the progression of kidney damage [16]. SGLT2i has been demonstrated to exert a reno-protective activity by ameliorating hyperfiltration and restoring tubule–glomerular feedback [17]. As a matter of fact, by promoting natriuresis and osmotic diuresis, they potentially enhance RAAS activation [18]. However, the association between SGLT2-i and systemic RAAS activation is not straightforward and additional studies are needed to clarify this relationship. Similarly, GLP-1 receptor agonists (GLP1-RA), a novel glucose lowering drug class, have been demonstrated to reduce angiotensin II levels and to inhibit their action via the post-receptor pathway. Moreover, a natriuretic effect was observed, probably mediated by the inhibition of Na+/H+ exchanger isoform 3 (NHE3) in the proximal tubule. These pathophysiological changes add up to the downregulation of pro-inflammatory and pro-fibrotic pathways, such as nuclear factor (NF)-κB and transforming growth factor-β (TGF-β) [19].

Angiotensin II (Ang II), the most powerful vasoconstrictor agent of RAAS, has been shown to play a key role in cell proliferation, hypertrophy, reactive oxygen species (ROS) generation, inflammation, and extracellular matrix (ECM) production through the induction of cytokines, chemokines, and growth factors [20]. Furthermore, Ang II mediates renal disorder by promoting the phenotypic switch of fibroblasts to myofibroblasts, which increases in the periglomerular and peritubular spaces contributing to ECM deposition [21]. This peptide participates in local and systemic hemodynamic regulation, and it mediates tubulointerstitial fibrosis [22] by stimulating the endogenous synthesis of TGF-β and connective tissue growth factor (CTGF). Interestingly, CTGF expression is blocked by the AT1R antagonist in vivo and in vitro experiments [23,24].

TGF-β acts on multiple cell types, and can induce renal fibrosis via activation of both canonical (Smad-based) and non-canonical (non-Smad-based) signalling pathways, which result in the activation of myofibroblasts, excessive ECM production, and inhibition of ECM degradation [25]. Through the Smad-based pathway, Smad3 can bind directly to Smad-binding elements within gene promoters to enhance transcription. Smad2 and Smad4 act as regulators of Smad3-based gene transcription [26]. TGF-β1 can also induce a fibrotic response through indirect mechanisms (non-Smad-based). On the other hand, TGF-β1 can induce the apoptosis of endothelial cells and podocytes [27], promoting glomerular and interstitial fibrosis. In addition, TGF-β1 is a potent inducer of a mesenchymal gene expression program that induces the transition of epithelial cells, endothelial cells, and intrinsic renal fibroblasts into α-smooth muscle actin (SMA)-expressing myofibroblasts [28]. A-SMA, as well as other cytoskeleton proteins, is demonstrated to be implicated in vascular cell migration and transition [29]. The detrimental, non-hemodynamic effects of RAAS activation on the kidney parenchyma are shown in Figure 1.

More recently, additional mechanisms that regulate the action of TGF-β1/Smad signalling in fibrosis were described, including short and long noncoding RNA molecules and epigenetic modifications of DNA and histone proteins [30]. A better understanding of the fibrotic pathways regulated by TGF-β has identified alternative therapeutic targets. MRAs, acting on mineralocorticoid receptors and antagonizing the action of aldosterone at mineralocorticoid receptors, have been suggested as an effective supplementary treatment for the existing clinical treatment of CKD cases. First, the non-steroidal selective mineralocorticoid receptor antagonist (MRA), finerenone [31], has demonstrated clinical benefits in CKD patients with type 2 diabetes, possibly through its direct anti-fibrotic effects.

Directly attenuating excess aldosterone production using selective aldosterone synthase inhibitors such as baxdrostat has successfully been used in patients with resistant hypertension, and is a promising approach for the progression of CKD. This may work differently than classical MRA because it efficiently blunts even ‘nongenomic’ and mineralocorticoid receptor-independent effects, which may be enhanced during treatment with MRA as they induce a rebound in aldosterone levels [32,33].

Another approach is the direct targeting of TGF-β1. Re-establishing the balance between profibrotic Smad3 activation and antifibrotic Smad7 action could constitute a novel approach to treating AH [30]. Among these drugs, pirfenidone, recently approved for idiopathic pulmonary fibrosis treatment, underwent a phase 2 trial for the treatment of kidney fibrosis [34].

Moreover, SGLT2i was shown to reduce Ang II-mediated kidney injury in mice [35] and had an inhibitory effect on TGF-β1 induced expression of thrombospondin-1 (THBS1), tenascin-C (TNC), and platelet-derived growth factor-β (PDGF-B), key mediators of interstitial fibrosis in human proximal tubular cells [36].

### 2.2. Endothelin-1 and Its Signaling Pathway

Endothelin (ET) appears to be increased in CKD as a response to hyperglycemia, hypertension, acidosis, and insulin or proinflammatory cytokines. Through activating the type A endothelin receptor (ETA), ET causes vasoconstriction of the efferent arterioles with hyperfiltration, podocyte damage with increased permeability and proteinuria, and GFR decline. Beyond the hemodynamic effects, ET-1 induced podocyte injury, mesangial cell proliferation, and mesangial matrix accumulation. In addition to these effects, ET-1 can induce inflammation and fibrosis [16], as the overexpression of ET-1 resulted in interstitial fibrosis in transgenic mice expressing human ET-1 that can be reversed only by ETA-selective receptor antagonists. Therefore, endothelin receptor antagonists (ERAs) have been proposed as a therapeutic strategy to reduce renal fibrosis, inflammation, and the progression of kidney disease. Moreover, ERAs may be particularly useful for treating salt-sensitive hypertension [37].

Nevertheless, most ERAs are still under investigation in ongoing clinical trials, or are not used in clinical practice because of a lack of efficacy or adverse events related to their use. Currently, apart from those approved to treat pulmonary arterial hypertension, atrasentan with an 1800-fold selectivity for ETA has been shown to reduce the risk of renal events and albuminuria in diabetic kidney disease, and is under study in an ongoing clinical trial in patients with proteinuric glomerular diseases.

The use of specific ETA receptor antagonists is to prevent edema. As the main deleterious effect related to ERAs is sodium retention with edema, their combination with gliflozins in the treatment of CKD has been proposed and is under investigation (i.e., zibotentan/dapagliflozin). Using a dual angiotensin II type 1 receptor/endothelin blocker (sparsentan) is expected to show additive renoprotective effects. It reduces blood pressure in hypertensive patients [28] and is studied in focal segmental glomerulosclerosis and IgA nephropathy patients. [38].

### 2.3. Innate Immunity Response and Oxidative Stress

In the past years, evidence has suggested that AH is, at least partially, an immune-mediated inflammatory disorder. AH is shown to be associated with the accumulation of immune cells into the kidneys, which impede vascular relaxation and enhance sodium reabsorption, mediating sodium sensitivity and tissue injury [39].

T cells, B cells, mast cells, macrophages, and dendritic cells (DCs) produce cytokines that are particularly relevant to AH. Monocytes and macrophages have been demonstrated to enhance vasoconstriction and sodium retention. DCs and B lymphocytes may stimulate the rise of BP by modulating T cell function. Furthermore, DCs produce mediators such as interleukin (IL)-1β and IL-6, which could modify BP independently of T cells [40,41].

In the presence of AH, the accumulation of activated Th17 cells producing the pro-inflammatory cytokine IL-17 has been demonstrated in several tissues, especially in high salt conditions. Firstly, IL-17 increases BP by inhibiting endothelial nitric oxide (NO) production, increasing ROS formation, and promoting fibrosis. Interestingly, by lowering the level of IL-17, better BP control is achieved and the inflammatory response is reduced [42]. Secondly, IL-17 has a major role in arterial stiffening, because of the increased synthesis and deposition of collagen it mediates within the vessel wall [43]. Thirdly, IL-17 can be considered a mediator of direct renal injury by enhancing renal sodium retention and glomerular injury [44]. As has been observed in a model of diabetic nephropathy, an IL-17 neutralizing antibody (IL-17A) improved kidney lesions and disease progression [45]. These data suggest that IL-17A may constitute a therapeutic target in the future.

Regarding innate immunity involvement, NOD-, LRR-, and pyrin domain-containing protein 3 (NLRP3) is the most well-recognized Nod-like receptor and the most widely studied inflammasome in the field of kidney disease [46]. The NLRP3 inflammasome is a cytosolic multiprotein caspase-activating complex platform involved in innate immunity required to mature and release IL-1β and IL-18 [47]. Activating the NLRP3 inflammasome requires a priming signal to induce the transcription of both NLRP3 and pro-IL-1β, and a second signal to prompt the oligomerization of the inflammasome. Several ligands can induce NLRP3 priming, including the TLR 2 ligand Pam3CSK4 (Pam3) and the TLR4 ligand LPS through the activation of NF-κB [48]. Several studies have discovered the roles and mechanisms of the NLRP3 inflammasome in AH, describing NLRP3 activation as a major mediator of inflammatory response via caspase-1 activation. The several functions of NLRP3 in regulating renal necro-inflammation and fibrosis emphasize the urgent need for specific NLRP3 inhibitors because of the broad therapeutic potential they offer for the treatment of CKD, in particular in the presence of AH. Interestingly, SGLT2i downregulates NLRP3 expression, ameliorating kidney fibrosis [49].

Of note, GLP1-RA alleviates oxidative stress in the kidney by reducing NADPH4 oxidase (NOX 4) expression and NADPH oxidase activity, decreasing renal expression of inducible nitric oxide synthase (iNOS) and cyclooxygenase-2 (COX-2), and by enhancing expression of catalase and glutathione peroxidase [19].

### 2.4. Hypoxia

Hypertension and kidney injury are associated with renal hypoxia. Hypoxic conditions are associated with the activation of RAAS, ROS production, and inflammatory response, and lead to the accumulation of ECM, resulting in fibrosis [50]. Moreover, low oxygen tension upregulates hypoxia-inducible factor (HIF), a transcriptional regulator influencing oxygen delivery and consumption [51]. HIF-1α activates the transcription of molecules that decrease renal oxygen consumption and promote a transition from oxidative metabolism to glycolysis, leading to the anaerobic production of adenosine triphosphate (ATP). HIF-2α is responsible for increasing erythropoietin transcription, leading to erythropoiesis in the kidney and liver, and vascular endothelial growth factor (VEGF) production, which plays a major role in angiogenesis. Together, the activation of HIF-1α and HIF-2α promotes the autophagic clearance of damaged mitochondria and peroxisomes by reducing metabolic demand and, consequently, oxygen consumption. HIF modulates inflammatory response to injury with the opposite effect as that of the isoform: HIF-1α activation is accompanied by the release of pro-inflammatory and pro-fibrotic molecules, while HIF-2α is linked to the suppression of inflammatory and fibrotic response [52].

Oxygen is essential for aerobic metabolism, and hypoxia may induce oxidative stress by increasing ROS production. Although receiving 20% of cardiac output, the kidney is highly sensitive to hypoxic injury due to medullary arterial–venous oxygen shunts [53]. Sodium reabsorption in the proximal tubule is an active process requiring large amounts of ATP. Oxygen is fundamental for the mitochondrial oxidative process, a pathway that permits the production of 30–36 molecules of ATP. Increased demand for oxygen due to high metabolic expenditure is usually balanced by increased blood flow. However, enhanced blood flow in the kidney implies an increased glomerular filtration rate (GFR) that, in turn, increases tubular sodium reabsorption. Conversely, oxidative stress activates the HIF pathway and induces endoplasmatic reticulum (ER) stress, which in turn promotes unfolded protein response (UPR) pathway-related apoptosis leading to kidney disease progression. Moreover, the increased production of ROS in endothelial cells reduces NO production and activity together with inflammatory molecules [53]. In a recent study, roxadustat, a stabilizer of HIFα family proteins currently used to treat CKD anaemia by promoting endogenous EPO synthesis, demonstrated a potent anti-hypertensive effect in Ang II-infused mice. This effect may be mediated through the upregulation of endothelial nitric oxide synthase (eNOS), modulation of AT-IIRs, and inhibition of oxidative stress, which concomitantly have been shown to prevent vascular thickening, cardiac hypertrophy, and kidney injury [53].

Recently, gliflozins raised strong interest for their proven anti-hypoxic effect; in fact, SGLT2i inhibiting proximal sodium–glucose cotransport might reduce hypoxia in the renal cortex by lowering O_2_ cortical consumption, leading to HIF-1α suppression; on the other hand, the activation of starvation sensors, namely 5′ AMP-activated protein kinase (AMPK) and Sirtuin1, leads to HIF-2α upregulation. Altogether, these pathways alleviate parenchymal inflammation and fibrosis [54]. Moreover, SGLT2i prevents mitochondrial ROS accumulation by enhancing non-shivering thermogenesis, promoting “metabolic flexibility” and eventually restoring autophagic flux by inhibiting the mammalian target of the rapamycin complex 1 (mTORC1) pathway [55].

### 2.5. Uric Acid

Uric acid (UA) is implicated in the vascular and systemic inflammatory response, contributing to the development and progression of both AH [56] and renal injury [57]. The research on the possible pathways involved is increasing because of the great impact of hyperuricemia on cardiovascular disease and mortality [58,59].

The increase in the prevalence of AH, diabetes, obesity, and renal disease, which has occurred over the past 100 years, has seen a parallel progressive increase in UA levels. Although the increase in UA may be due, at least in part, to impaired renal function, the association between UA and cardiovascular events appears to be independent of serum creatinine [60]. The pathogenetic mechanisms through which UA could induce AH and kidney damage are various (Figure 2).

Microcrystalline nephropathy is the first mechanism of UA that has been shown to affect the kidney. Recent studies have interestingly shown a hyperechoic pattern of Malpighi pyramids in gouty patients. The pattern was independently associated with coronary heart disease, arterial hypertension, hyperuricemia, and decreased estimated glomerular filtration rate [61], suggesting monosodium urate microcrystals could affect vessels and kidneys.

Hyperuricemia also causes damage to the preglomerular vessels, affecting the afferent arterioles’ autoregulatory capacity, resulting in glomerular hypertension. Lumen narrowing due to arteriolar wall thickening also results in renal hypoperfusion, with ischemia and decreased GFR, both possible mechanisms through which UA could cause systemic hypertension and renal parenchymal damage [57].

Microcrystalline nephropathy of gout could be an important target for urate-lowering treatment (ULT) [62]. Nevertheless, the debate regarding the efficacy of ULT for kidney protection is ongoing [63]. The inconsistency in the results regarding its utility in kidney protection is probably due to population selection biases [64]. There are UA hyperproducers and under-execrators, with the latter being more frequently affected by kidney damage [65].

Recent experimental studies have shown that the internalization of UA increases oxidative stress within cells [66]. In human epithelial tubular cells, UA causes increased free radicals and oxidative stress via NADPH oxidase and induces inflammation [67,68]. Moreover, it decreases NO at both plasma and cellular levels with endothelial dysfunction and activation of the RAAS [69]. At this point, two phases take over: an acute probably reversible phase, with thickening of the vessel leading to the development of AH; a chronic phase in which the vessel wall is thickened and renal damage is now established. Some authors have sustained that UA overproducers may benefit more from ULT, as it may target intracellular UA [65].

Animal studies have highlighted the ability of UA to increase sodium reabsorption through the dysfunction of ENaC channels, increasing their expression in a dose-dependent manner, and thus suggesting a new causal mechanism of sodium-sensitive in conditions of hyperuricemia. An additional pathogenetic mechanism of sodium-sensitive hypertension involving the pro-renin receptor has been demonstrated in a model of rats fed a high amount of fructose. UA has been demonstrated as a major mediator of this pathway, leading to intrarenal RAAS activation and increased sodium absorption in the kidney [70]. Compelling in vitro evidence that UA is a key player in vascular remodelling by the proteasome and RAAS activation has been recently reported. In these experiments, UA treatment induces changes in vascular smooth muscle cells (VSMC) typical of initial phases of atherosclerosis with a structural and functional transition into the proliferative/migratory phenotype of VSMC with an increase in the cellular migration rate. Interestingly, these changes are abolished by the block of internalization of UA and inhibition of the proteasome, RAAS, or MEK activity [71].

Once the damage has been developed, an effective protective result of UA modulation is harder to demonstrate, which might justify the lack of strong favourable data about the efficacy of urate lowering treatment in CV protection [72]. As a partial solution to this failure, based on the reported findings, inhibition of ubiquitin–proteasome system might serve as a therapeutic target in managing vascular damage, and RAAS inhibition might be thought to prevent UA-induced atherosclerosis, even in the absence of AH. Several studies have demonstrated that SGLT2i reduces UA levels by enhancing its elimination; this effect seems to be mediated by competition for uric transporters glucose transporter 9 (GLUT9) and urate transporter 1 (URAT1) [73,74]. Moreover, the AMPK pathway activated by SGLT2i promotes upregulation of transporter ATP-binding cassette super-family G member 2 (ABCG2), with a subsequent UA lowering effect [75].

### 2.6. Ageing

Hypertension accelerates and accentuates renal aging. The kidney experiences progressive functional decline, as well as macroscopic and microscopic histological alterations [76], and molecular pathway activation in response to systemic comorbidities such as AH.

In aging, due to various intrinsic and extrinsic stimuli throughout life, cells can stop dividing, acquiring a “senescent” phenotype. They appear enlarged and flattened with multiple nuclei and vacuoles. Senescent cells are viable, metabolically altered, and resistant to apoptosis. In addition, they develop a proinflammatory senescence-associated secretory phenotype (SASP) exerting detrimental effects on nearby cells. Markers of senescence are senescence associated β galactosidase (SA-β gal) activity, upregulation of cell cycle inhibitors such as p16Ink4a (p16), p53 and p21, activin A, lamin B1, γ-H2 Histone family member X, and telomere shortening [77].

Klotho plays an important role in maintaining endothelial integrity [78], in angiogenesis and vasculogenesis, which are impaired in Klotho mutant mice, a model of typical aging [79]. Low levels of Klotho have been found in essential and in renovascular hypertension and they directly correlate with GFR, suggesting the involvement of Klotho in the pathogenesis of kidney injury in hypertensive disorders [80]. The decrease in Klotho expression and its circulating levels have been shown to be associated with aging [81]. Cellular senescence has been implicated in the development of renal diseases, and thus tubular, glomerular, stromal, and vascular cells can assume senescent features. In hypertensive kidney disease, senescence can negatively affect renal function and increases the frequency of end-stage renal disease in older people [82]. Growing evidence indicates that oxidative stress [83], RAAS, and, consequently, the inflammatory response, play a key role in aging kidneys [84].

The longevity phenotype is characterized by an increased number of mitochondria and through upregulation of the pro-survival genes nicotinamide phosphoribosyltransferase (Nampt) and sirtuin 3 (Sirt3) in the kidney [85]. Therefore, the epigenetic modifications controlled by Sirtuins could be responsible for longevity, supporting the possibility of their beneficial role in the context of age-associated diseases. Moreover, it has been shown that Sirtuin 6 (SIRT6) has an important role in preventing endothelial dysfunction, a key player in AH and its complications. Indeed, in desoxycorticosterone acetate/salt–induced and Ang II–induced hypertensive mice, SIRT6 overexpression reduces BP, improves endothelium-dependent vasorelaxation, and ameliorates renal injury. SIRT6 prevents and delays progressive CKD induced by hyperglycaemia, kidney senescence, AH, and lipid accumulation by regulating mitochondrial biogenesis, and has antioxidant, anti-inflammatory, and antifibrosis effects. Lastly, SIRT 1, SIRT 3, and SIRT 6 are well-known nutrient deprivation sensors, and their expression is augmented by SGLT2i administration [86].

### 2.7. miRNA

The latest findings about the role of vascular remodelling in organ damage mediated by AH regard miRNA, small single-stranded non-coding RNA molecules appearing to regulate gene expression. Cheng et al. [87] have demonstrated that endogenous miR-204, a highly expressed miRNA in the kidney, has a prominent role in safeguarding the kidneys against common causes of CKD, namely diabetes and AH. The authors sustained miR-204 attenuated renal injury due to the mitigation of interlobular artery thickening, as shown in multiple mice models. As interlobular artery thickening is the most common pathologic finding in patients with AH-induced renal injury, we could suggest that the protective effect of miR-204 on renal interlobular arteries may involve the attenuation of vascular remodelling, contrasting renal tissue hypoxia as the disease progresses. Moreover, lademirsen, a miR-21 antagonist, has been demonstrated to reduce inflammation and kidney fibrosis in animal models of hypertensive and diabetic nephropathy. There are ongoing phase I and II clinical trials to ascertain its beneficial effect on Alport disease [35,88]. To modulate molecular regulatory networks that ultimately underlie the development and progression of chronic renal injury, it is important to investigate the role of regulation by miRNA of other target genes or pathways.

## 3. Histopathological Kidney Changes during Arterial Hypertension

Among hypertensive patients with CKD, increased systolic BP, reduced diastolic BP, and elevation in pulse pressure (PP) are typical features. This pattern of isolated systolic hypertension is indicative of arterial remodelling in CKD, characterized by premature vascular ageing and accelerated arterial stiffening [89]. We have linked the classic picture of hypertensive nephrosclerosis involving podocyte damage and loss and tubulointerstitial fibrosis with the latest knowledge, which includes histopathology and molecular mechanisms inducing vascular remodelling. In particular, we describe recent knowledge about phenotype switching, the ageing process, and the newest role of micro-RNA (miRNA). More recently, the presence of a specific subtype of nephropathy associated with congestion has been suggested. Even though no diagnostic criteria have been established, and no renal-specific histological patterns have been reported, studies regarding this issue may help to improve the handling and therapeutic principles in affected patients.

### 3.1. Smooth Muscle Cell Phenotypic Switching

Vascular smooth muscle cells are the predominant cell type controlling blood vessel stiffness and BP. They switch between alternate phenotypes of contractile in non-pathological settings to the synthetic–proliferative phenotype, driving cardiovascular disease. Ahmed et al. [90] investigated the role of VSMC under physiological conditions and blood vessel physiology, describing how arterial stiffening could be transmitted to the microcirculation of other organs. The authors describe how different molecules and structures result in the transition between contractile vs. the synthetic–proliferative VSMC phenotype, through the mechanisms that involve cytoskeletal proteins, myosin light chains (MLC-20 and MLC-17), and myosin isoforms. The complex crosstalk between VSMCs and their surrounding matrix in healthy and in pathological conditions provide new insights into the mechanisms that regulate the phenotypic switch [71].

A recent in vitro study reporting uric acid as a key player in cytoskeleton changes of VSMCs demonstrated how proteasome inhibition, angiotensin receptor blockers, and the blockade of uric acid intracellular internalization prevented the development of dysfunctional VSMC, providing new potential therapeutic targets contrasting VSMCs cytoskeleton changes and vascular remodelling [90]. Animal studies also reported a beneficial effect of Angiotensin 1-7 that resulted in the inhibition of mitochondrial fragmentation, ROS generation, and hyperproliferation in murine models. Due to the prevented Ang II-induced vascular remodelling, the authors suggested that enhancing Ang 1-7 actions may provide a novel therapeutic strategy to prevent or delay VSMC switching [91].

### 3.2. Epithelial-to-Mesenchymal Transition and Endothelial-to-Mesenchymal Transition

Hypertension-induced epithelial-to-mesenchymal transition EMT is a major mechanism of renal fibrosis due to the activation and accumulation of ECM-producing fibroblasts or myofibroblasts [92]. Moreover, during the atherosclerotic processes, endothelial-to-mesenchymal transition (EndMT) links disturbed shear stress and inflammation, with tissue remodelling, promoting plaque formation [93]. EndMT has recently been suggested to promote fibrosis and is recognized as a novel mechanism generating myofibroblasts. Similar to EMT, during EndMT, endothelial cells lose their adhesion and apical-basal polarity to form highly invasive, migratory, spindle-shaped, elongated mesenchymal cells. More importantly, biochemical changes accompany these distinct changes in cell polarity and morphology, including the decreased expression of endothelial markers and the acquisition of mesenchymal markers [94]. Recent data suggest that AdipoRon, an orally active synthetic adiponectin receptor agonist, could be a potential therapeutic option to prevent renal fibrosis by attenuating EMT in hypertensive patients [93].

### 3.3. Renal Congestion

Renal dysfunction in heart failure (HF) has traditionally been considered to result from decreased renal perfusion and associated neural and hormonal changes. Recently, persistent venous congestion was identified as a major contributor [95].

Histopathological lesions reported in rodent models of renal congestion include enlargement of peritubular capillaries, pericyte detachment, and tubular injury—mainly tubular atrophy and fibrosis [96].

Timely treatment with the restoration of venous pressure might resolve congestion-induced renal dysfunction. With this aim, using tolvaptan in adjunction to a sequential nephron blockade might be very effective. Moreover, in a small cohort of ADPKD patients, treatment with tolvaptan is associated with reduced oxidative stress [97].

## 4. Conclusions

High BP is well recognized to play a key role in the pathogenesis of kidney damage occurrence and progression. While clinical data suggest that failure to achieve adequate BP control likely contributes to suboptimal outcomes in kidney disease, we believe that more directly interfering with the molecular mechanisms associated with the development of AH and end-organ damage might significantly reduce kidney damage and thereby improve the overall outcome of patients with AH.

In this review, we discussed the molecular interactions in the pathophysiology of AH and showed how they affect the kidney in a particular way. Considering the currently available data that we tried to summarize, we hope that an improved understanding of the alterations in the molecular pathways involved in the development of AH and the associated renal damage may lead to developing new therapeutic strategies targeting the more downstream mediators.

## Figures and Tables

**Figure 1 ijms-24-09422-f001:**
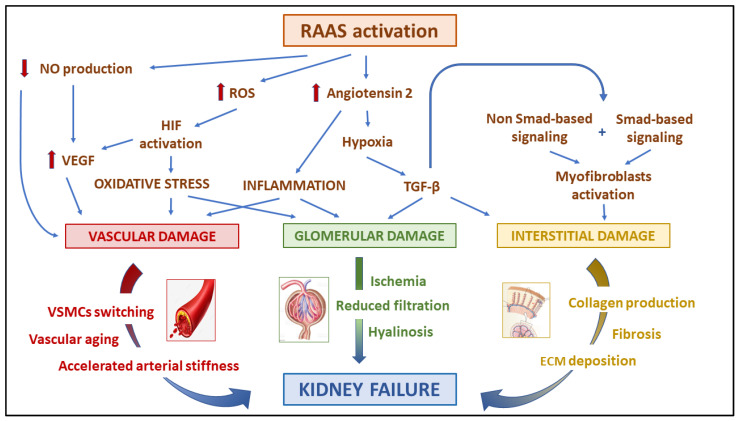
Non-haemodynamic effects of RAAS activation on the kidney parenchyma. RAAS activation enhances vascular and glomerular inflammation increasing angiotensin II production and ROS generation; altogether, these pathways cause hypoxia, oxidative stress, and HIF activation. Moreover, NO downregulation increases VEGF production, leading to VSMC phenotype switch, resulting in vascular aging and accelerating arterial stiffness. On the other hand, TGF-β, via Smad and non-Smad-based signalling, induces myofibroblasts activation, collagen production, ECM deposition, and, eventually, interstitial fibrosis and glomerular damage. ECM, extracellular matrix; HIF, hypoxia inducible factor; NO, nitric oxide; RAAS, renin angiotensin aldosterone system; ROS, reactive oxygen species; TGF- β, transforming growth factor-β; UA, uric acid; VEGF, vascular endothelial growth factor; VSMC, vascular smooth muscle cell.

**Figure 2 ijms-24-09422-f002:**
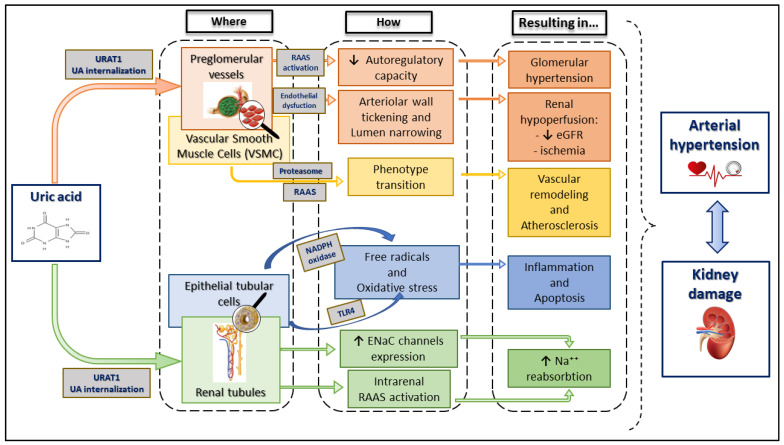
Uric acid effects on the kidney parenchyma determining arterial hypertension and kidney damage. UA internalization via URAT1 increases glomerular pressure by activating RAAS; renal hypoperfusion is strictly related to vascular lumen narrowing and arteriolar wall thickening promoted by UA-induced endothelial dysfunction. VSMC transition into proliferative/migratory phenotype enhances vascular remodelling via proteasome and RAAS stimulation. On the other hand, UA induces the inflammation and apoptosis of epithelial tubular cells through ROS production and increased oxidative stress. Moreover, intrarenal RAAS activation and ENaC upregulation lead to increased Na+ reabsorption, contributing to hypertension physiopathology. eGFR, estimated glomerular filtration rate; ENaC, epithelial sodium channel; NADPH, nicotinamide adenine dinucleotide phosphate; RAAS, renin–angiotensin–aldosterone system; TLR4, toll-like receptor 4; UA, uric acid; URAT1, urate transporter 1.

**Table 1 ijms-24-09422-t001:** Promising therapeutic approaches contrasting arterial hypertension-induced kidney damage.

Mechanisms by Which ah Affects the Kidney	Emerging Therapies Targeting Specific Kidney Pathways
1. **RAAS** Angiotensin II overproductionCell proliferation and hypertrophyROS generationInflammationECM productionTGF- β/Smad signalling	 Non-steroidal, selective MRA (i.e., Finerenone)  Aldosterone synthasi inhibitor (i.e., Baxdrostat)  AT1 antagonist  Direct targeting of TGF-β1 (i.e., Pirfenidone)  SGLT2i  GLP1-RA
2. **Innate immunity response/oxidative stress** Cytokine IL-17 overproductionInhibition of endothelial NO productionActivation of NLRP3 inflammasome	 IL-17 neutralizing antibody  Specific NLRP3 inhibitors  SGLT2i
3. **Endothelin-1** Podocyte injury,Mesangial matrix accumulation,FibrosisInflammationCell proliferation and hypertrophy	 Selective type A ERA (i.e., Atrasentan)  Dual angiotensin-II type 1/endothelin receptor blockers (i.e., Sparsentan)  Combination of ERAs with SGLT2i (i.e., Zibotentan/dapagliflozin)
4. **Hypoxia** HIF upregulationVEGF overexpressionEndoplasmic reticulum stress	 Stabilizer of HIFα family  SGLT2i
5. **Uric acid** Oxidative stress/inflammationRAAS activationENaC channels dysfunctionProteasome activationVSMCs phenotype switching	 Urate lowering treatment  Inhibition of ubiquitin-proteasome system  Angiotensin receptor blockers  Blockade of UA internalization (i.e., Losartan)  SGLT2i
6. **Histopathological kidney changes** VSMCs switchingEMT and EndMTAgeingmiRNA	 Adiponectin receptor agonist (i.e., AdipoRon)  Target: Angiotensin 1-7  Target: Sirtuins (SIRT 1,3,6)  miR-204  Antagonist targeting miR-21(i.e., Lademirsen)

AT, angiotensin receptor; ECM, extracellular matrix; EMT, epithelial-to-mesenchymal transition; EndMT, endothelial-to-mesenchymal transition; ERA, endothelin receptor antagonist; GLP1-RA, glucagon like peptide-1 receptor antagonist; HIF, hypoxia inducible factor; IL, interleukin; MRA, mineralocorticoid receptor antagonist; NO, nitric oxide; RAAS, renin angiotensin aldosterone system; ROS, reactive oxygen species; SGLT2i: sodium-glucose cotransporters inhibitors; TGF-β, transforming growth factor-β; UA, uric acid; VEGF, vascular endothelial growth factor; VSMC, vascular smooth muscle cell.

## Data Availability

Not applicable.

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
