# Peer review of "Non-Haemodynamic Mechanisms Underlying Hypertension-Associated Damage in Target Kidney Components"

_ijms, 2023, doi:10.3390/ijms24119422_

Round 1

Reviewer 1 Report

Reviewer comments and suggestions

The authors in this study focussed on the interactions of pathways linking the molecular consequences of Arterial hypertension (AH) to kidney damage, suggesting how old and new therapies could attempt to protect the kidney. The authors discussed antihypertensive treatment to limit the damage of the kidney such as renin angiotensin aldosterone system (RAAS) inhibitors, gliflozins, endothelin receptor antagonists and their combination, the kidney damage associated to AH be far from being defeated. Additionally, they discussed the recent studies on molecular mechanisms of AH-induced kidney damage have identified novel potential therapeutic targets. 

Overall, the manuscript looks weak to be accepted in this journal. However, a few concerns/comments needed to be explained/modified. 

  1. The contrast word did not work in line number 14
  2. Why the introduction section was italic, is there was any specific reason for this
  3. Comments for table 1 The table should be professional, not looking to be added in the manuscript. Please modify
  4. Line 74-76 Seems grammatical error in the sentence
  5. Line 79 Please name the class of drug “gliflozin” and then explain it
  6. A few of the words were in bold format, kindly change into appropriate 
  7. Line 122 The authors need to discuss completely if they added a new type of drug or molecules
  8. Section 2.2 Please modify the title of this section if possible
  9. Line 253-256 The authors need to discuss the section with the help of citing figure 2 (also the figure needs reference citation) from where authors prepared this diagram

10. Comment for section 2.5 It’s not the mechanism 

related to "Kidney injury in arterial hypertension" better to differentiate it

11. The manuscript needs a through revision and it lacks the section that was included in the conclusion part. The author can add more mechanisms to valid their manuscript in a better way.

12. Please check the guidelines of mdpi, it seems that the authors need to modify all the references

Extensive editing of English language required.

Reviewer 2 Report

The review by Russo and colleagues is an elegant narrative review on the mechanisms of hypertension-induced kidney damage. I commend the Authors for their nice work and I just add few comments:

- Although results are not that strong yet, I would at least cite the potential role of endothelin receptor antagonist, which are just mentioned in the abstract but not in the manuscript (PMID: 36893968)

-  Although used in autonomic dominant polycystic kidney disease, vasopressin antagonists may play a role in the future in nephroprotection from other diseases

- As for uric acid, I would add few important concepts: 

      1) inconsistency in current evidence regarding the role of hyperuricemia and its treatment in nephroprotection are also due to population selection biases (PMID: 36815099). In fact, there are UA hyperproducers and underexcretors, the latter being more frequently affected by kidney failure. Hyperproducers may benefit more from xanthine oxidase inhibitors, since it may target intracellular UA, which is a key player in UA-induced cellular damage, as the Authors correctly stated. Recent reviews summarise current evidence on nephroprotective properties of different UA-lowering drugs (PMID: 36999206). This reviewer believes UA-lowering drugs should be added to Table 1 in the UA section. 

2) An important mechanism of UA-induced kidney damage is microcrystalline nephropathy (also called gouty nephropathy). Interestingly, it has been found that renal ultrasound may detect patients with microcrystalline nephropathy, allowing for a more specific population selection in large trials (PMID: 32898570, PMID: 37060437). 

Round 2

Reviewer 1 Report

No more comments

Author Response

We thank the Reviewer for his/her accuracy in his/her work.